# The Impact of Cancer-Related Fatigue on HRQOL in Survivors of Childhood Cancer: A DCCSS LATER Study

**DOI:** 10.3390/cancers14122851

**Published:** 2022-06-09

**Authors:** Adriaan Penson, Iris Walraven, Ewald Bronkhorst, Heleen Maurice-Stam, Martha A. Grootenhuis, Margriet Van der Heiden-van der Loo, Wim J. E. Tissing, Helena J. H. Van der Pal, Andrica C. H. De Vries, Dorine Bresters, Cécile Ronckers, Marry M. Van den Heuvel, Sebastian J. C. M. M. Neggers, Birgitta A. B. Versluys, Marloes Louwerens, Saskia M. F. Pluijm, Leontien C. M. Kremer, Nicole Blijlevens, Eline Van Dulmen-den Broeder, Hans Knoop, Jacqueline Loonen

**Affiliations:** 1Department of Hematology, Radboud University Medical Center, Geert-Grooteplein Zuid 8, 6500 HB Nijmegen, The Netherlands; jacqueline.loonen@radboudumc.nl; 2Department for Health Evidence, Radboud Institute for Health Sciences, Radboud University Medical Center, Nijmegen, Geert-Grooteplein 21, 6500 HB Nijmegen, The Netherlands; iris.walraven@radboudumc.nl (I.W.); ewald.bronkhorst@radboudumc.nl (E.B.); nicole.blijlevens@radboudumc.nl (N.B.); 3Princess Máxima Center for Pediatric Oncology, Heidelberglaan 25, 3584 CS Utrecht, The Netherlands; h.maurice-stam@prinsesmaximacentrum.nl (H.M.-S.); m.vanderheiden@prinsesmaximacentrum.nl (M.V.d.H.-v.d.L.); w.j.e.tissing@umcg.nl (W.J.E.T.); h.j.h.vanderpal@prinsesmaximacentrum.nl (H.J.H.V.d.P.); d.bresters@prinsesmaximacentrum.nl (D.B.); cecile.ronckers@uni-oldenburg.de (C.R.); m.m.vandenheuvel-eibrink@prinsesmaximacentrum.nl (M.M.V.d.H.); s.neggers@erasmusmc.nl (S.J.C.M.M.N.); a.b.versluijs@prinsesmaximacentrum.nl (B.A.B.V.); s.m.f.pluijm@prinsesmaximacentrum.nl (S.M.F.P.); l.c.m.kremer@prinsesmaximacentrum.nl (L.C.M.K.); 4Department of Psychology, Princess Máxima Center for Pediatric Oncology, Heidelberglaan 25, 3584 CS Utrecht, The Netherlands; m.a.grootenhuis-2@prinsesmaximacentrum.nl; 5Department of Pediatric Oncology/Hematology, University of Groningen/University Medical Center Groningen, Hanzeplein 1, 9713 GZ Groningen, The Netherlands; 6Department of Pediatric Oncology, Erasmus Medical Center, Doctor Molewaterplein 40, 3015 GD Rotterdam, The Netherlands; a.c.h.devries@erasmusmc.nl; 7Willem-Alexander Children’s Hospital, Department of Pediatrics, Leiden University Medical Center, Einthovenweg 20, 2333 ZC Leiden, The Netherlands; 8Department of Health Services Research, Carl von Ossietzky University of Oldenburg, Ammerländer Heerstraβe 114, 26129 Oldenburg, Germany; 9Department of Pediatric Oncology, Erasmus Medical Center—Sophia Children’s Hospital, Doctor Molewaterplein 40, 3015 GD Rotterdam, The Netherlands; 10Department of Medicine, Section Endocrinology, Erasmus Medical Center, Doctor Molewaterplein 40, 3015 GD Rotterdam, The Netherlands; 11Leiden University Medical Center, Department of Internal Medicine, Albinusdreef 2, 2333 ZA Leiden, The Netherlands; m.louwerens@lumc.nl; 12University Medical Center Utrecht, Wilhelmina’s Children’s Hospital, Lundlaan 6, 3584 EA Utrecht, The Netherlands; 13Department Pediatric Oncology, Emma Children’s Hospital, University of Amsterdam, Meibergdreef 9, 1105 AZ Amsterdam, The Netherlands; 14Department of Pediatric Oncology/Hematology, Amsterdam University Medical Center, Meibergdreef 9, 1105 AZ Amsterdam, The Netherlands; e.vandulmen-denbroeder@amsterdamumc.nl; 15Department of Medical Psychology, Amsterdam University Medical Centers, University of Amsterdam, Amsterdam Public Health Research Institute, Meibergdreef 9, 1105 AZ Amsterdam, The Netherlands; hans.knoop@amsterdamumc.nl

**Keywords:** cancer survivorship, fatigue, health-related quality of life

## Abstract

**Simple Summary:**

Survivors of childhood cancer have an increased risk to experience symptoms of severe and persistent fatigue. We studied how fatigue might affect the health-related quality of life of these survivors. Questionnaire items asking about a broad range of daily life aspects were compared between fatigued survivors, survivors without fatigue and the general Dutch population. A total of eleven aspects were studied which were all negatively affected by fatigue, with the largest impact seen for Vitality (how much energy does a person have), General Health (perception of current and future health) and Role Limitations (work-related activities). Results show the negative impact fatigue can have on the daily lives of survivors and why it is important to treat fatigue adequately.

**Abstract:**

Background: Early detection and management of late effects of treatment and their impact on health-related quality of life (HRQOL) has become a key goal of childhood cancer survivorship care. One of the most prevalent late effects is chronic fatigue (CF). The current study aimed to investigate the association between CF and HRQOL in a nationwide cohort of CCS. Methods: Participants were included from the Dutch Childhood Cancer Survivor Study (DCCSS) LATER cohort, a nationwide cohort of CCS. Participants completed the Checklist Individual Strength (CIS) to indicate CF (CIS fatigue severity subscale ≥ 35 and duration of symptoms ≥6 months) and the Short Form-36 (SF-36) and TNO (Netherlands Organization for Applied Scientific Research) and AZL (Leiden University Medical Centre) Adult’s Health-Related Quality of Life questionnaire (TAAQOL) as measures for HRQOL. Differences in mean HRQOL domain scores between CF and non-CF participants were investigated using independent samples *t*-tests and ANCOVA to adjust for age and sex. The association between CF and impaired HRQOL (scoring ≥ 2 SD below the population norm) was investigated using logistic regression analyses, adjusting for confounders. Results: A total of 1695 participants were included in the study. Mean HRQOL domain scores were significantly lower in participants with CF. In addition, CF was associated with impaired HRQOL on all of the domains (except physical functioning) with adjusted odds ratios ranging from 2.1 (95% CI 1.3–3.4; sexuality domain) to 30.4 (95% CI 16.4–56.2; vitality domain). Conclusions: CF is associated with impaired HRQOL, urging for the screening and regular monitoring of fatigue, and developing possible preventative programs and interventions.

## 1. Introduction

With a growing population of childhood cancer survivors (CCS) [1,2,3], early detection and management of the late effects of treatment and their impact on daily life has become a key goal of survivorship care [4]. Personalized cancer survivorship care [5], aimed at empowering survivors and supporting self-management, using Patient Reported Outcomes (PROs) to evaluate late effects, will become more and more important. PROs capture issues affecting quality of life that matter to the patient, for example, the ability to work, participate in social activities, practice sports and perform household activities or chores. Investigating PROs and what affects them is relevant when aiming to improve quality of life in CCS. A core concept of PROs are Health-Related Quality of Life (HRQOL) outcomes, reflecting the subjective perception of health [6,7].

Here we use the term HRQOL, referring to a person’s subjective appraisal of physical, mental and social well-being, matching the 1947 World Health Organization’s (WHO) definition of ‘health’ [8]. Previous studies showed CCS to have impaired HRQOL compared to the general population [9,10]. Several childhood diagnosis and sociodemographic factors were associated with impaired HRQOL in CCS [9,10,11]. In addition, late effects, such as cardiovascular or pulmonary dysfunction, were associated with poor HRQOL [12,13]. 

A late effect often reported by CCS is chronic fatigue (CF) [14], indicating severe fatigue which persists for 6 months or longer. Previous studies showed fatigue to negatively affect HRQOL in CCS [15,16,17,18], but only included subgroups of CCS or did not take into account the severity and/or persistence of fatigue symptoms. In addition, other possible confounding factors that were related to poor HRQOL (for example, depression or having a medical condition [19,20]) were not taken into account. Therefore the association of CF (indicated with a validated cut-off point and including duration of symptoms) with HRQOL remains unclear. In the current study, we aim to overcome these limitations and determine the association between CF and HRQOL in CCS, after correcting for confounders. 

Thus, the aim of the current study was to investigate the independent association of CF and HRQOL in a nationwide cohort of adult CCS including all childhood malignancies. We believe it is important to assess the association between CF and HRQOL, as such new evidence will improve our understanding of the role of CF in decreasing HRQOL in CCS and in determining whether CF could potentially be a feasible factor to target when aiming to improve HRQOL. 

## 2. Methods

### 2.1. Design & Participants

Cross-sectional data were collected for the Dutch Childhood Cancer Survivor Study (DCCSS) LATER Fatigue Study [21] as part of the DCCSS LATER 2 study (Feijen, Teepen, Loonen et al., under review). Participants aged ≥18 years were included from the DCCSS LATER cohort [22], a nationwide cohort of CCS diagnosed before the age of 18 between 1 January 1963 and 31 December 2001 in the Netherlands, and who are at least five years post diagnosis. All participants who were able to read and speak Dutch and who gave written informed consent to participate received an invitation by mail to visit the outpatient clinic for care and participation in clinical research between 2017 and 2020 (details described elsewhere [21]). If eligible survivors did not respond within a few weeks, a reminder was sent via mail. Data on childhood cancer diagnosis and treatment were collected by data managers using a uniform and standardized protocol [23]. Data on fatigue status was collected with questionnaires during the clinic visit and questionnaires assessing HRQOL were completed at home (on paper or digitally). The DCCSS LATER fatigue study was approved by the Medical Research Ethics Committee of the Amsterdam University Medical Centers (registered at toetsingonline.nl, NL34983.018.10).

### 2.2. Measures

#### 2.2.1. Fatigue

The Checklist Individual Strength (CIS) [24], a 20-item questionnaire, scored on a 7-point Likert Scale, was used to assess fatigue severity. The CIS was designed to measure several aspects of fatigue with the subscales fatigue severity (eight items), concentration (five items), motivation (four items) and physical activity level (three items). The CIS is a reliable and valid instrument for the assessment of fatigue, with a score of 35 or higher on the CIS fatigue severity subscale (range 8–56) indicating severe fatigue [25]. The psychometric properties of the CIS were shown to be good in the CCS group (high correlation with other fatigue measures and four-factor structure confirmed, with all factors having high internal consistency) [26]. Symptom duration was asked in a separate item. To identify participants experiencing CF, we defined CF as severe fatigue, indicated with a score of 35 or higher on the CIS fatigue severity subscale [25], which persisted for at least six months [27].

#### 2.2.2. HRQOL

The 36-item Short Form Health Survey (SF-36) [28,29] was used to determine eight HRQOL domains (see Appendix A). For each domain, item scores are coded, summed and transformed to a scale from 0 (worst) to 100 (best) following instructions described elsewhere [28,30]. The survey was constructed for self-administration by persons 14 years of age and older. The Dutch version of the SF-36 was shown to be valid and reliable (item internal consistency and -discriminant validity as well as known groups comparisons met criterium values) [31].

The TNO (Netherlands Organization for Applied Scientific Research) and AZL (Leiden University Medical Centre) Questionnaire for Adult’s Quality of Life (TAAQOL) [32] subscales, sleep (SL), sexuality (SE) and cognitive functioning (CO) were used in the current study. Scale scores were calculated and linearly transformed to a 0–100 scale (following instructions described elsewhere [33]) with higher scores indicating better functioning. The questionnaire was validated in both the general population as well as in patients with chronic diseases, confirming the assumed questionnaire structure, with all of the subscales having high reliability [32,34].

The HRQOL domains, physical functioning (PF), role-physical (RP), bodily pain (BP), general health (GH), vitality (VT), social functioning (SF), role-emotional (RE) and mental health (MH) of the SF-36 and the domains, sleep (SL), sexuality (SE) and cognitive functioning (CO) of the TAAQOL were investigated in the current study (see Appendix A for details). Participants who scored ≥2 standard deviations from the general population mean [31,33] on a HRQOL subscale were identified as “impaired” for this domain.

#### 2.2.3. Other Measures

The following questionnaires were completed as measures for possible confounders, i.e., depression, anxiety, sleep quality, somatic comorbidities and sociodemographic factors, based on their relation with fatigue and HRQOL in previous literature [10,12,13,14,20,35,36,37].

The Hospital Anxiety and Depression Scale (HADS) [38] was used to assess symptoms of anxiety and depression. The HADS assesses anxious and depressive feelings over the past four weeks, with both subscales containing seven items on a 4-point Likert scale. A cutoff score of ≥8 for both the anxiety subscale and the depression subscale was used to identify (sub)clinical cases [39].

The Pittsburg Sleep Quality Index (PSQI) [40] was used to assess overall sleep quality. The PSQI, with a total of 18 items (four free response items and fourteen 4-point Likert scale items), generates a total of seven component scores, namely subjective sleep quality, sleep latency, sleep duration, habitual sleep efficiency, sleep disturbances, use of sleeping medication and daytime function. The components are scored 0–3, with a total score ranging from 0–21 and higher scores indicating poorer sleep. A total score >5 was used to indicate poor sleep [40].

A general health questionnaire, containing items to assess demographic characteristics (age, sex, employment status and education level) was completed. Details about this questionnaire are described elsewhere [21]. In addition, almost all of the participants in the current study participated in the 2013 LATER questionnaire study (DCCS LATER 1 study; Teepen, Kok, Feijen et al. under review) assessing physical health issues (*n* = 1367). These self-reported health issues were validated based on self-reported medication use and medical files when needed and were used to categorize participants as having 0, 1–2, or >2 clinically relevant somatic comorbidities, based on a previously published outcomes set [41].

### 2.3. Statistical Analysis

To examine possible selection bias between study participants and non-participants (eligible CCS that did not return informed consent or did not complete study questionnaires), Chi-Square tests (with Cramér’s V as effect size) were calculated to compare the groups on sex, decade of birth, childhood cancer diagnosis, decade of diagnosis, treatment with chemotherapy and /or radiotherapy (yes/no). 

Participants were assigned to one of the following groups: a) CCS without CF (NCF group); b) CCS with CF (CF group). If one CIS fatigue severity subscale item was missing, this was imputed with the mean value of the remaining seven CIS fatigue severity subscale items. To identify significant differences in mean total scores of the HRQOL domains between the CF group and NCF group, independent samples *t*-tests were calculated (with Cohen’s d effect size) and an ANCOVA was completed, with age and sex as the covariates to correct for differences between the groups. Mean total score differences between the CF group and population norms [31,33] were tested with independent samples *t*-tests. 

To investigate the association of CF with HRQOL, univariate and multivariable logistic regression was performed, allowing to adjust for potential confounders. Univariate logistic regression was completed with the HRQOL domains as dependent variable (impaired yes/no) and CF (yes/no) as independent variable. Multivariable logistic regression was completed to determine whether a possible association would remain after adjustment for confounders (age, sex, BMI, employment status, educational level, sleep quality, depression, anxiety, number of somatic comorbidities, childhood diagnosis and treatment; see Appendix A). Missing data of the independent variables (Appendix A; Little’s MCAR test *p* = 0.34) were imputed using multiple imputation (five imputed datasets, using Rubin’s rules to pool the analyses) [42,43,44]. Variance inflation factors (VIF) were calculated for all of the independent variables with a threshold of >10 to test for multicollinearity [45]. IBM SPSS (IBM Corp. Released 2017. IBM SPSS Statistics for Windows, Version 25.0. Armonk, NY, USA: IBM Corp) was used for the statistical analyses.

## 3. Results

### 3.1. Participants

A total of 2282 CCS participated in the DCCSS LATER 2 study (48.2% of eligible persons), of whom 1695 completed the fatigue and HRQOL questionnaires for the current study, so that fatigue status (CIS fatigue severity subscale score and fatigue duration) and at least one of the eleven HRQOL subscale scores could be calculated (74.3%; flowchart in Figure 1).

Participants differed from non-participants on sex and received childhood cancer treatment, however the effect sizes were small (Cramér’s V ranged 0.03–0.12; Appendix A). A total of 744 persons declined participation and were therefore excluded from all analyses.

Participant characteristics are shown in Table 1. Compared to CCS without CF, CCS with CF were more often female (63.4% vs. 43.6%), aged ≥40 years (35.0% vs. 29.7%), diagnosed after 1990 less often (48.6% vs. 54.4%) and received only chemotherapy less often (47.5% vs. 56.1%), but more often a combination of radiotherapy and chemotherapy (38.6% vs. 32.0%).

### 3.2. Chronic Fatigue and HRQOL Scale Scores

CCS with CF scored significantly lower on all of the HRQOL domains compared to CCS without CF (Figure 2). Independent *t*-tests resulted in *p*-Values < 0.001 for all domains, also after adjustment for age and sex (ANCOVA *p*-Values < 0.001), with large Cohen’s d effect sizes (>0.8) for all of the domains except SE (0.59). Mean differences ranged from 14.0 (SE) to 41.6 (RP) with the largest differences seen on the domains RP, RE and VT (mean difference >30). In addition, CCS with CF scored below the population norm values on all of the domains (*p* < 0.001).

### 3.3. Association Chronic Fatigue with Impaired HRQOL

Univariate logistic regression showed CF to be associated with impaired HRQOL domains (Table 2). The ORs for CF were significantly increased for all of the HRQOL domains, with the ORs for RP, BP, MH, VT GH and SF > 10. After adjustment for confounders, CF remained significantly associated with impaired HRQOL (Table 2), except for the PF domain (*p* = 0.069). The largest OR was seen for VT, but also the domains RP, BP and GH showed ORs exceeding five-fold risks for CCS with CF, compared to CCS without CF. 

## 4. Discussion

The aim of the current study was to investigate the association of CF with HRQOL in a nationwide cohort of CCS. Compared to CCS without CF and the general population, CCS with CF scored markedly lower on almost all of the studied HRQOL domains, independent of other potentially influential factors, demonstrating CF to be associated with worse HRQOL in CCS. These results emphasize the importance of including CF screening and monitoring in survivorship care aiming to improve quality of life, as it was shown to affect the daily lives of CCS on multiple aspects. 

Recent studies conducted in the DCCSS LATER cohort already showed CCS to more often have impaired HRQOL than the general population on several domains [9,10]. The current study focused on the role of CF on impaired HRQOL in CCS. Except for PF, all of the HRQOL domains showed a clear and independent association with CF, after adjustment for many known characteristics. The domain PF illustrates a person’s ability to perform physically demanding (household) tasks, such as walking the stairs or doing groceries. In the unadjusted analysis, the association between PF and CF was less strong than for the other domains, suggesting that PF is less affected by CF in CCS. In addition, factors other than fatigue might be more associated with this HRQOL domain. Having other health issues and poor sleep were previously related to decreased physical functioning in other patient populations, for example, survivors of breast cancer [46,47], and therefore it is possible that these factors in particular might have caused the effect of CF to flatten in the current study. The strongest association was seen between CF and the VT domain (OR 30.352). This strong association could be expected as feelings of vitality are no doubt affected by fatigue. However, the CIS fatigue severity subscale and SF-36 VT subscale had a moderate correlation (data not shown) indicating both scales to reflect different concepts.

Previous studies show CF to have a negative impact on HRQOL in multiple patient populations [48,49,50,51], and also, in subgroups of CCS, it was already suggested that (chronic) fatigue affects HRQOL [15,17]. The current study confirms this, showing the impact of CF on a broad range of HRQOL subscales in a generalizable cohort of CCS. Studies including CCS of all childhood diagnoses (except CNS tumors, in Frederick et al.) [16,18] also showed the fatigued CCS to have decreased HRQOL domains compared to the non-fatigued CCS, however, the questionnaires they used to assess fatigue symptoms did not have validated cut-off scores to indicate severe fatigue and the duration of symptoms was not taken into account. In addition, 44% of the participants in the study by Frederick et al. [18] were aged 12–19 years, and >52% of the participants in the study by Mulrooney et al. [16] were Hodgkin lymphoma survivors, whilst the current study focused on long-term adult CCS (aged >18 years) of all childhood malignancies. The current study combined the strengths of previous studies (cohort including all childhood diagnoses, questionnaire with validated cut-off score to indicate severe fatigue, take into account symptom duration) and showed CF to negatively impact HRQOL domains in a generalizable cohort CCS. Our results show CF to play an important role in decreasing multiple HRQOL aspects in CCS, emphasizing that it should be addressed in CCS care when aiming to improve HRQOL.

Owing to the cross-sectional design of the study, we do not know if CF causes HRQOL to decrease. It is also plausible that HRQOL is causally related to the occurrence and/or duration of CF. Therefore, although our study shows a strong association between CF and HRQOL, the causal relation between CF and impaired HRQOL remains to be studied, preferably in a longitudinal study. In addition, differences in mean HRQOL scores between the CF group and the general population were not adjusted for age and sex, as we did not have sufficient data of the general population to do so. This should be taken into account when interpreting the HRQOL domain differences between these groups. Another limitation of the current study was the comparatively low number of participants with domain specific HRQOL scores defined as ‘impaired’ for the purpose of the analyses (ranging from 46 (PF and BP) to 321 (CO)). Including multiple independent variables in the multivariable logistic regression analyses with few ‘cases’ could have affected the power to detect associations. However, independent variables were included one-by-one until the final model was analyzed so that power issues concerning the models (large OR confidence intervals, for example) that may occur would be noticed (which was not the case). Furthermore, data on self-reported somatic comorbidities, here used as a summary frequency score, were collected in 2013 (Teepen, Kok, Feijen et al., under review) which is slightly earlier than the parameters studied here (2016–2020). Although it is possible that new conditions might have affected some survivors (e.g., false negatives or too low counts in the current data), false-positives are unlikely since the questionnaire focused on chronic conditions, likely still present at the time of the current study. 

An international Guideline for Childhood, Adolescent and Young Adult Cancer Survivors that was published in 2020 already stressed the importance of screening regularly for fatigue and of treating it adequately [52]. The current study adds to this call, as it indicates CF as a late effect clearly impairing HRQOL on various domains. Early detection of severe fatigue symptoms and providing a (personalized) intervention could prevent symptoms from becoming worse and affecting HRQOL. Using a screening instrument, for example, the Short Fatigue Questionnaire (SFQ) [53], could help to detect severe fatigue early. The SFQ is a 4-item questionnaire and is fast and easy to administer with a validated cut-off score to indicate severe fatigue [26,54]. To get a more complete assessment of fatigue and its impact, a multidimensional fatigue questionnaire, such as the CIS, and a HRQOL questionnaire, such as the SF-36 or TAAQOL, could be used. Completing these questionnaires would probably take less than 20 min and scoring could be automated using online assessment. Psychosocial therapies, such as cognitive behavioral therapy or exercise therapy, could be possible interventions to consider, as previous studies have shown it to be adequate for treating CF in survivors of adult cancer and its potential was also shown in CCS, although more studies are needed to confirm these results [52,55,56,57,58].

To conclude, early detection and management of the late effects of treatment and their impact on quality of life have become a key goal of CCS health care. Understanding the impact of specific late effects on HRQOL is crucial when aiming to improve the quality of CCS daily lives. The current study shows CF to have a negative impact on multiple HRQOL domains, indicating the urge for structural screening and, when needed, adequate treatment of fatigue symptoms in CCS care.

## Figures and Tables

**Figure 1 cancers-14-02851-f001:**
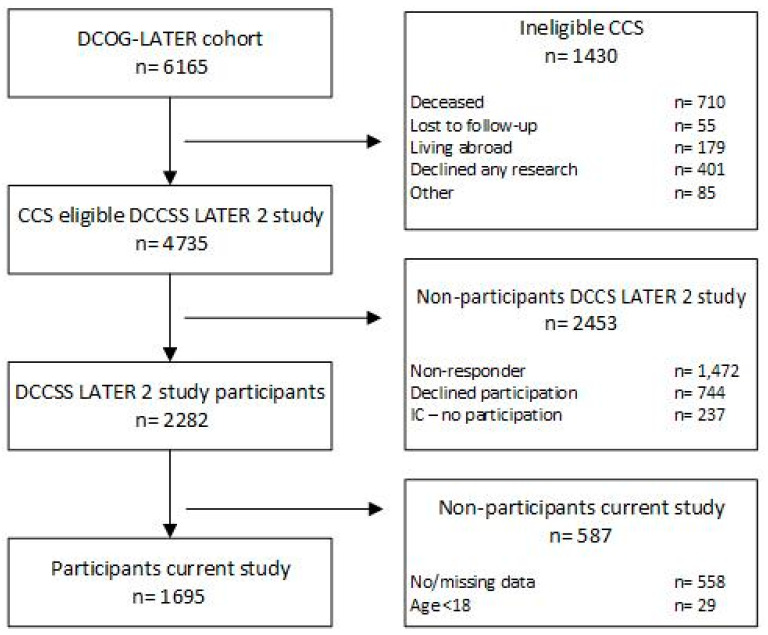
Flowchart of CCS participants. IC—no participation: Did return informed consent wanting to participate but did not participate (due to logistic reasons or lack of time for example). No/missing data: no or incomplete data for the CIS fatigue severity subscale, duration of fatigue symptoms or all of the HRQOL subscales.

**Figure 2 cancers-14-02851-f002:**
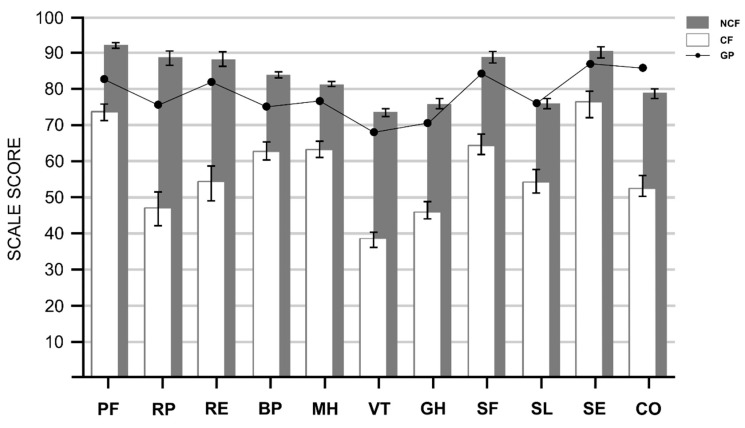
Mean total scores for HRQOL domains for CCS with and without chronic fatigue. Mean Health Related Quality of Life (HRQOL) subscale scores of Childhood Cancer Survivors (CCS) with chronic fatigue (CF) and CCS without chronic fatigue (NCF). Error bars show 95% Confidence Interval. Black line represents mean subscale scores of the General Population (GP) [31,33]. Subscales of SF-36: PF = Physical Functioning; RP = Role physical; RE = Role Emotional; BP = Bodily Pain; MH = Mental Health; VT = Vitality; GH = General Health. Subscales of TAAQOL: SL = Sleep; SE = Sexuality; CO = Cognitive Functioning.

**Table 1 cancers-14-02851-t001:** Participant Characteristics for total cohort and by chronic fatigue status.

Characteristic	Total Cohort CCS (*n* = 1695)	CCS NCF (*n* = 1304)	CCS CF (*n* = 391)	*p*-Value ^e^
	N (%)	N (%)	N (%)	
Sex				<0.001
Male	878 (51.8)	735 (56.4)	143 (36.6)	
Female	817 (48.2)	569 (43.6)	248 (63.4)	
Age at assessment (years)				0.050
<20	30 (1.8)	26 (2.0)	4 (1.0)	
20–29	485 (28.6)	390 (29.9)	95 (24.3)	
30–39	656 (38.7)	501 (38.4)	155 (39.6)	
≥40	525 (30.9)	387 (29.7)	137 (35.0)	
Age at diagnosis (years)				0.262
0–5	770 (45.4)	605 (46.4)	165 (42.2)	
5–10	458 (27.0)	350 (26.8)	108 (27.6)	
10–15	370 (21.8)	272 (20.9)	98 (25.1)	
15–18	97 (5.7)	77 (5.9)	20 (5.1)	
Primary childhood cancer diagnosis ^a^				0.259
Leukemia	581 (34.3)	459 (35.2)	122 (31.2)	
Non-Hodgkin lymphoma ^b^	210 (12.4)	165 (12.7)	45 (11.5)	
Hodgkin lymphoma	121 (7.1)	95 (7.3)	26 (6.6)	
CNS	158 (9.3)	114 (8.7)	44 (11.3)	
Neuroblastoma	97 (5.7)	70 (5.4)	27 (6.9)	
Retinoblastoma	8 (0.5)	5 (0.4)	3 (0.8)	
Renal tumors	193 (11.4)	150 (11.5)	43 (11.0)	
Hepatic tumors	17 (1.0)	16 (1.2)	1 (0.3)	
Bone tumors	101 (6.0)	76 (5.8)	25 (6.4)	
Soft tissue tumors	124 (7.3)	88 (6.7)	36 (9.2)	
Germ cell tumors	56 (3.3)	46 (3.5)	10 (2.6)	
Other and unspecified ^c^	29 (1.7)	20 (1.5)	9 (2.3)	
Period of childhood cancer diagnosis				0.002
1963–1969	28 (1.7)	21 (1.6)	7 (1.8)	
1970–1979	226 (13.3)	151 (11.6)	75 (19.2)	
1980–1989	542 (32.0)	423 (32.4)	119 (30.4)	
>1990	899 (53.0)	709 (54.4)	190 (48.6)	
Childhood cancer treatment ^d^				0.047
Surgery only	109 (6.4)	81 (6.2)	28 (7.2)	
Chemotherapy, no radiotherapy	917 (54.1)	731 (56.1)	186 (47.6)	
Radiotherapy, no chemotherapy	93 (5.5)	68 (5.2)	25 (6.4)	
Radiotherapy and chemotherapy	568 (33.5)	417 (32.0)	151 (38.6)	
No treatment/treatment unknown	8 (0.5)	7 (0.5)	1 (0.3)	
Recurrence				0.460
No	1468 (86.6)	1125 (86.3)	343 (87.7)	
Yes	227 (13.4)	179 (13.7)	48 (12.3)	

Abbreviations: CCS = Childhood Cancer Survivors; NCF = group without chronic fatigue; CF = group with chronic fatigue; CNS = Central Nervous System. ^a^ Diagnostic groups included all malignancies covered by the third edition of the International Classification of Childhood Cancer (ICCC-3) as well as multifocal Langerhans cell histiocytosis; ^b^ Includes all morphology codes specified in the ICCC-3 under lymphomas and reticuloendothelial neoplasms, except for Hodgkin lymphomas. Includes multifocal Langerhans cell histiocytosis; ^c^ Includes all morphology codes specified in the ICC-3 under other malignant epithelial neoplasms and malignant melanomas and other and unspecified malignant neoplasms; ^d^ Treatment data included primary treatment and all recurrences; ^e^ Chi square test for differences between NCF and CF group.

**Table 2 cancers-14-02851-t002:** Association of CF with impaired HRQOL.

HRQOL Subscale	Unadjusted OR	95% CI	Adjusted OR ^a^	95% CI	Adjusted OR ^b^	95% CI
SF-36						
PF	4.24	3.34–7.66	3.68	2.01–6.72	2.01	0.95–4.26
RP	11.47	7.99–16.46	10.16	7.05–14.66	6.34	4.19–9.56
RE	9.13	6.58–12.67	8.81	6.30–12.30	4.01	2.67–6.04
BP	10.42	5.34–20.33	8.52	4.33–16.76	5.72	2.64–12.39
MH	12.58	6.53–24.26	14.92	7.58–29.34	2.47	1.04–5.84
VT	49.66	28.63–86.13	52.69	30.03–92.44	30.35	16.41–56.16
GH	13.06	8.11–21.03	12.05	7.43–19.55	6.88	3.94–12.01
SF	10.84	6.94–16.93	10.59	6.72–16.71	3.87	2.26–6.63
TAAQOL						
SL	5.02	3.58–7.03	4.56	3.23–6.43	n/a *	n/a *
SE	4.44	2.99–6.59	4.14	2.77–6.19	2.08	1.28–3.38
CO	6.29	4.84–8.19	6.07	4.63–7.94	3.06	2.21–4.23

Results of multivariable logistic regression analyses with HRQOL domains as dependent outcome variable and CF as independent variable. ^a^ Adjusted for age and sex; ^b^ Adjusted for age, sex, BMI, employment status, educational level, poor sleep quality, anxiety, depression, number of somatic comorbidities, childhood cancer diagnosis, childhood cancer treatment. * The HRQOL domain sleep was excluded from this analysis as sleep quality was considered a confounder and both factors show too much overlap to include in the same model. Abbreviations: HRQOL = Health related quality of life; CF = chronic fatigue; OR = Odds ratio; 95% CI = 95% confidence interval; PF = Physical Functioning; RP = Role physical; RE = Role Emotional; BP = Bodily Pain; MH = Mental Health; VT = Vitality; GH = General Health; SL = Sleep; SE = Sexuality; CO = Cognitive Functioning.

## Data Availability

The data underlying this article were provided by the DCOG-LATER Consortium under license. Data will be shared on request to the corresponding author with permission of the DCOG-LATER consortium.

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
