# Peer review of "The Impact of Cancer-Related Fatigue on HRQOL in Survivors of Childhood Cancer: A DCCSS LATER Study"

_cancers, 2022, doi:10.3390/cancers14122851_

Round 1
Reviewer 1 Report
I have several questions and comments of this paper. It does address a very important topic in childhood cancer.
1. Are there several more updated papers that are less than 6-8 years old? I noted several in the introduction in the first 24-26 references that were older papers.
2. Under measures, do you more appropriate terms to describe the reliability of the measures rather than good. What does that mean to the reader? This would apply to all the measures in this paper.
3. Under methods, how did the participants receive an invitation? What method of continued recruitment did you use? I suspect it is similar to the CCSS.
4. In the discussion section, I recommend the term 'focused' rather than 'zoomed' in the role of CF.
Author Response
Response to reviewers
We thank all the reviewers for their time to review our manuscript entitled ‘The impact of cancer-related fatigue on HRQOL in survivors of childhood cancer; a DCCSS LATER study’. We found their suggestions very helpful and have in response to their comments, added additional information to the revised manuscript (marked in yellow). Below, we respond to every comment point by point, showing our answers in red.
Reviewer 1
I have several questions and comments of this paper. It does address a very important topic in childhood cancer.
- Are there several more updated papers that are less than 6-8 years old? I noted several in the introduction in the first 24-26 references that were older papers.
We thank the reviewer for the suggestion to update the references in the introduction. We agree that several references were outdated, and have therefore, when possible, replaced some old references with newer ones.
Removed:
Bosetti, C., et al., (2010) European Journal of Cancer
Ghatta, G., et al., (2002) European Journal of Cancer
Reulen, R.C., et al., (2007) Int J Cancer
Han, K.T., et al. (2014) Health Qual Life Outcomes
Merchant, J.A., et al. (2014) J Occup Environ Med
Edgerton, J., Roberts L., Below, S.V., (2008) Cancer Epidemiol Biomarkers Prev
Included:
Armstrong G.T., et al., (2016) New England Journal of Medicine
Malvezzi, M., et al., (2021) Cancer
Van Gorp, M. et al., (2021) Cancer
Van Erp, L.M.E., et al., (2021) European Journal of Cancer
Hult, M., (2020) Social Indicators Research
- Under measures, do you more appropriate terms to describe the reliability of the measures rather than good. What does that mean to the reader? This would apply to all the measures in this paper.
We thank the reviewer for the comment and have added some statements to in more detail describe the measures of psychometric properties of the outcome measures. We have added these statements to the revised manuscript as follows:
Fatigue measure CIS
“Psychometric properties of the CIS were shown to be good in CCS (high correlation with other fatigue measures and four-factor structure confirmed with all factors having high internal consistency).”
HRQOL measures
SF-36
“The Dutch version of the SF-36 was shown to be valid and reliable (item internal consistency and -discriminant validity as well as known groups comparisons met criterium values).”
TAAQOL
“The questionnaire has been validated in both the general population as well as in patients with chronic diseases, confirming the assumed questionnaire structure, with all subscales having high reliability”
- Under methods, how did the participants receive an invitation? What method of continued recruitment did you use? I suspect it is similar to the CCSS.
Thank you for the suggestion. We have now added to the original recruitment statement that eligible survivors received an invitation by mail and we also included a statement that non-responders received a reminder by mail after a few weeks. It is now written as follows in the revised manuscript:
“All participants who were able to read and speak Dutch and who gave written informed consent to participate received an invitation by mail to visit the outpatient clinic for care and participation in clinical research between 2017 and 2020 (details described elsewhere). If eligible survivors did not respond within a few weeks, a reminder was sent via mail.”
- In the discussion section, I recommend the term 'focused' rather than 'zoomed' in the role of CF.
Thank you for the suggestion. We have changed the term ‘zoomed’ into ‘focused’.
Reviewer 2 Report
The Authors present a paper: "The impact of cancer-related fatigue on HRQOL in survivors of childhood cancer; a DCCSS LATER study" very interesting from a psychosocial point of view.
Introduction, Methods, Statistical analysis, and Discussion are very well reported and developed.
Tables and Figures are appropriate and explicative.
Only a minor remark: I ask the Authors to add a short paragraph of conclusions where they have to report how to suggest minimal requirements on how to study chronic fatigue and test for HRQOL, and adding the mean-time to conclude this test.
Author Response
Response to reviewers
We thank all the reviewers for their time to review our manuscript entitled ‘The impact of cancer-related fatigue on HRQOL in survivors of childhood cancer; a DCCSS LATER study’. We found their suggestions very helpful and have in response to their comments, added additional information to the revised manuscript (marked in yellow). Below, we respond to every comment point by point, showing our answers in red.
Reviewer 2
The Authors present a paper: "The impact of cancer-related fatigue on HRQOL in survivors of childhood cancer; a DCCSS LATER study" very interesting from a psychosocial point of view.
Introduction, Methods, Statistical analysis, and Discussion are very well reported and developed.
Tables and Figures are appropriate and explicative.
Only a minor remark: I ask the Authors to add a short paragraph of conclusions where they have to report how to suggest minimal requirements on how to study chronic fatigue and test for HRQOL, and adding the mean-time to conclude this test.
We thank the reviewer for the suggestion and have added the following to the second last paragraph of the discussion:
“Using a screening instrument, for example the Short Fatigue Questionnaire (SFQ) [54], could help to detect severe fatigue early. The SFQ is a 4-item questionnaire and is fast and easy to administer with a validated cut-off score to indicate severe fatigue [27, 55]. To get a more complete assessment of fatigue and its impact, a multidimensional fatigue questionnaire such as the CIS and a HRQOL questionnaire such as the SF-36 or TAAQOL could be used. Completing these questionnaires would probably take less than 15 minutes and scoring could be automated using online assessment”
Reviewer 3 Report
The study of Penson et al. describes the impact of cancer-related fatigue on HRQOL in survivors of childhood cancer. This study is part of a large DCCSS LATER study, which represents a large unique national cohort of childhood cancer survivors. This study is well-written, and has an appropriate design. The results are interesting and relevant.
My most important comment is the following:
It is not new that fatigue does influence quality of life. Although this study describes the quality of life of childhood cancer survivors in more detail, the authors could improve their manuscript by focusing more on what is new in the current study, compared to the already existing data from their own studies and from studies from other international groups. Why does this study adds to the already existing knowledge?
Minor points:
Introduction:
The introduction is quite long and may be improved by being more concise to focus on CF.
Results:
Table 1 could be improved by adding the information about the differences between the two groups (NCF versus CF).
It seems that the lay-out of Table 1 is not correct. It seems that one primary childhood cancer diagnosis is missing. It is not clear to me where the number 581 (34.3) relates to. Please also have a look at the lay-out of the cancer treatment.
Author Response
We thank all the reviewers for their time to review our manuscript entitled ‘The impact of cancer-related fatigue on HRQOL in survivors of childhood cancer; a DCCSS LATER study’. We found their suggestions very helpful and have in response to their comments, added additional information to the revised manuscript (marked in yellow). Below, we respond to every comment point by point, showing our answers in red.
Reviewer 3
The study of Penson et al. describes the impact of cancer-related fatigue on HRQOL in survivors of childhood cancer. This study is part of a large DCCSS LATER study, which represents a large unique national cohort of childhood cancer survivors. This study is well-written, and has an appropriate design. The results are interesting and relevant.
My most important comment is the following:
It is not new that fatigue does influence quality of life. Although this study describes the quality of life of childhood cancer survivors in more detail, the authors could improve their manuscript by focusing more on what is new in the current study, compared to the already existing data from their own studies and from studies from other international groups. Why does this study adds to the already existing knowledge?
We thank the reviewer for the comment and agree that previous studies already showed the relation between fatigue and HRQOL. However, as stated in de introduction and discussion, these previous studies differed in methodology as they only focused on subgroups of CCS (certain diagnoses or age groups), did not include the duration of fatigue symptoms (thus not studying chronic fatigue in particular) and did not correct for multiple confounders. The current study aims to overcome these limitations in order to better determine the independent relation of chronic fatigue with HRQOL. To underline this, in agreement with the suggestion of the reviewer, we added the following sentence to the second last paragraph of the introduction: “In the current study we aim to overcome these limitations and determine the association between CF and HRQOL in CCS after correcting for confounders.”
Minor points:
Introduction:
The introduction is quite long and may be improved by being more concise to focus on CF.
We agree with the reviewer that the introduction is quite long and could be improved by focusing on chronic fatigue. We have therefore excluded the term cancer-related fatigue from the manuscript and focused on chronic fatigue. In addition, we have removed the following sentences (reducing the number of words from 501 to 397):
First paragraph, first sentence: “Survival of childhood cancer has improved significantly over the past decades”
Third paragraph, second sentence: “The National Comprehensive Cancer Network (NCCN) has defined CRF as a distressing, persistent, subjective sense of physical, emotional, and/or cognitive tiredness or exhaustion related to cancer and/or cancer treatment that is not proportional to recent activity and interferes with usual functioning.”
Third paragraph, last sentence: “By taking into account the severity and persistency of fatigue, CF closely matches the NCCN definition of CRF”
Results:
Table 1 could be improved by adding the information about the differences between the two groups (NCF versus CF).
Thank you for the suggestion to add information about differences between the groups. We have now added p-values of the chi square test for differences between the CF and NCF groups in Table 1.
It seems that the lay-out of Table 1 is not correct. It seems that one primary childhood cancer diagnosis is missing. It is not clear to me where the number 581 (34.3) relates to. Please also have a look at the lay-out of the cancer treatment.
Thank you for your detailed eye. We have changed the lay-out of the Table.